# Perspectives of the Elderly with Mild Cognitive Impairment Living Alone on Participating in a Dementia Prevention Program: A Q Methodology Study

**DOI:** 10.3390/ijerph17217712

**Published:** 2020-10-22

**Authors:** Li-Ting Lu, Chiu-Mieh Huang, Su-Fei Huang, Shu-I Wu, Jong-Long Guo

**Affiliations:** 1Department of Nursing, University of Kang Ning, Taipei 114, Taiwan; lyteen69@gmail.com; 2Department of Health Promotion and Health Education, National Taiwan Normal University, Taipei 106, Taiwan; 3Institute of Clinical Nursing, School of Nursing, National Yang-Ming University, Taipei 112, Taiwan; cmhuang@ym.edu.tw; 4Department of Senior Citizen Service, Mackay Junior College of Medicine, Nursing, and Management, Taipei 112, Taiwan; s331@mail.mkc.edu.tw; 5Xinyi District Health Center, Taipei City 110, Taiwan; shuyi4843@gmail.com

**Keywords:** community-dwelling, elderly living alone, mild cognitive impairment, Q methodology, dementia prevention

## Abstract

This study aimed to identify and describe the various patterns of perspectives among older adults with mild cognitive impairment (MCI) living alone on participating in a dementia prevention program. Q methodology was applied to investigate the perspectives of 30 community-dwelling elderly people with MCI living alone from March to August 2018. As Q methodology applies a forced distribution through the Q-sorting technique, it could capture participants’ perspective patterns. Thirty-two Q-statements were constructed to explore the participants’ attitudes regarding their participation in a dementia prevention program. The participants performed Q-sorting to rank the 32 statements into a Q-sort grid. Principal component analysis was conducted using the PQ Method 2.35 software to identify patterns in participants’ perspectives. Four patterns of shared perspectives, accounting for 54.65% of the total variance, were identified: (a) awareness of health benefits and readiness to take preventive actions; (b) emphasis on cost consideration, and not ready to participate; (c) concern about family’s attitude and needing family support; (d) emphasis on medical care and needing providers’ recommendation. The exploration of clusters of the elderly with MCI could assist health professionals in acknowledging elderly people’s attitudes and responses towards participating in a dementia prevention program.

## 1. Introduction

With the development of medical technology and public health, the average human life expectancy has been extended. In Taiwan, the proportion of older adults over 65 years of age was over 14.56% in 2018 [1]. According to the statistics of the Taiwan Ministry of Health and Welfare, there were 45,035 elderly living alone registered for care in 2019, accounting for 1.2% of the population aged 65 and over [2]. Elderly people living alone have been identified as a population with multiple health risks [3,4]. If elderly people living alone can receive appropriate assistance and care from their surrounding children, friends, and neighbors, living alone may not result in problems when they can live independently. However, if the elderly people live alone in poor health and cannot receive appropriate assistance and care, a health crisis may result [5]. 

In addition, one of the most common problems of community-dwelling elders living alone is a lack of interpersonal interaction. A previous study indicated that low social participation, less frequent social contact, and more loneliness were statistically significantly associated with the incidence of dementia [6]. Elderly people living alone may not detect changes in their cognitive status, so they cannot receive a timely dementia prevention program when their initial symptoms appear. Thus, elderly people living alone with early dementia symptoms may lose valuable opportunities for early detection and treatment. The International Dementia Association released the Global Dementia Report in 2015, which estimates that the number of people with dementia will reach 131.5 million by 2050. According to the report, the peak incidence of dementia in Europe and the United States is 80–89 years, and in Asia it is 75–84 years. Thus, older adults in Asia need to pay attention to dementia earlier in their lifespan compared to their counterparts in Europe and the US. 

According to statistics in Taiwan, mild cognitive impairment (MCI) is estimated to affect 18.16% of older adults over the age of 65, and 7.78% of the population is diagnosed with dementia [7]. Dementia is a disease rather than a normal aging phenomenon, and patients experience a combination of clinical symptoms associated with cognitive decline. Patients with dementia may have decreased brain function, memory loss, learning ability, social ability, cognitive dysfunction, an increasing need for assisted self-care, and problems in self-care [8]. Dementia patients’ attention, language ability, abstract thinking ability, sense of space, numeracy, judgment, and behavioral emotion control ability will be influenced, and they are prone to symptoms such as delirium, hallucinations, and interference behavior. These symptoms influence the severity of problems experienced in work, social relationships, and daily life functions [9]. 

A previous review article of clinical trials indicated that non-pharmacologic interventions can delay the progression of functional impairment or disability among community-dwelling dementia patients [10]. A UK program entitled “Thinking Fit” for dementia prevention in older adults with mild cognitive impairment that includes physical activities, group-based cognitive stimulation activities, and personal forms of cognitive stimulation activities was shown to reduce the risk of dementia [11]. In Japan, researchers have designed a complex dementia prevention program to improve the cognitive function of older adults that includes the components of body movement, brain training, nutrition education, and group learning activities. The study found that these interventions improved the cognitive function of the elderly [12]. 

The Taiwan government has promoted the long-term care 2.0 program since 2017, and dementia prevention and care is a critical service. The target population of the services is older adults with dementia or who are at high risk. The service aims to provide a dementia prevention program located in long-term facilities and community stations. The goal of the program is to maintain the physical, cognitive, and interpersonal functions of patients with MCI or dementia and effectively reduce the burden on family caregivers. For example, the program may consist of a series of training including computing skills, dance abilities, diet and nutrition skills, problem solving skills, and interpersonal interaction [13].

An effective dementia prevention program needs a successful marketing strategy to attract the targeted population. In the past, social marketing emphasizing the “4Ps”—product, place, price, and promotion—has been used in public health programs to achieve behavioral change [14]. However, the traditional 4P strategy does not focus on the concerns of program participants, and the participant-centered 4Cs (customer, cost, convenience, and communication), proposed by market scientist Lauterborn [15], could be used to market a dementia prevention program for high-risk elderly. The 4Cs emphasize that manufacturers should produce products that meet consumer needs, set prices at the cost consumers are willing to pay, consider the convenience of consumer shopping, and provide consumers with a two-way communication channel. Consumer demand will influence the purchase of a product, and consumers will buy in because they believe that the product has the ability to solve their problems [16]. Mindful of the 4Cs, our study aimed to elucidate participants’ perspective patterns on participating in a dementia prevention program, and their concerns are therefore a critical topic [17] in the study. Moreover, a previous study indicated the influence of the community also to be a critical determinant of older adults’ participation in health promotion programs [17]. Thus, the 4Cs could be supplemented with participants’ concern (**concern**) and the influence of the community (**community**) to become the 6C framework in the current study. The 6C framework in our study underscores the importance of participants’ needs, including (1) concern: emphasizing that participants will have increased interest in a health program because of their concerns; (2) consumer: meeting participants’ needs is more important than health program functions; (3) cost: participants are willing to pay to meet their own needs—e.g., monetary expenditure, time, physical and energy consumption, and risk-taking; (4) convenience: the marketing channel focuses on the convenience of participants when they intend to participate in a health program; (5) communication: a two-way communication between the program deliverer and participants needs to be emphasized to enhance mutual understanding and achieve a feasible marketable path; and (6) community: the influence of significant others on the participant [16]. We used the 6C framework to explore the patterns of perspectives among elderly people with MCI, living alone, on participation in a dementia prevention program. 

Q methodology integrates qualitative and quantitative research methods by combining conceptual categorization with the quantification of participants’ patterned perspectives [18,19], and it can identify the heterogeneity of participants’ subjectivity through the process of “Q-sorting”. Older adults would be able to disclose their perspectives about the dementia prevention program freely, without the constraints imposed by quantitative survey methods. The Q methodology has been widely applied in health topic research, such as ascertaining employees’ attitudes toward the resource demands of implementing the Baby-Friendly Hospital Initiative among maternity staff [20], patterns of treatment expectation and physician–patient relationships perceived by women receiving traditional Chinese medicine [21], single older adults’ differing perspectives on new relationships [22], patterns of parents’ perspectives on avoiding secondhand smoke exposure [23], older adults’ perspectives on fall-prevention beliefs [17], testing the usability of digital educational games for encouraging smoking cessation [24], and interaction between students and supervisory staff in drug use prevention [25].

Thus, the study aimed to identify and describe the various patterns of perspectives of elderly with mild cognitive impairment living alone on participating in a dementia prevention program. 

## 2. Method

### 2.1. Development of Q-statements

In order to develop Q-statements based on the 6C framework, the interview questions (Table 1) were sent to 30 participants prior to their face-to-face interviews, allowing the interviewees’ time to prepare. Each interview lasted between 2 and 3 h, was audio-recorded, and was subsequently transcribed verbatim. All the interviews were conducted in participants’ residence and in their native language. After the interview, 162 statements were extracted from the verbatim transcripts. Following a thorough review by the research team, these were condensed into 32 statements. Three health professionals were invited to examine the content validity, and two previously interviewed participants were recruited to test the face validity of the Q-statements. Ambiguous and confusing language was modified to ensure that the interviewees could comprehend all the statements. 

### 2.2. Participants

The study was conducted in a district of Taipei City, Taiwan, in 2018, where 41,622 elderly people were residing at the time. In total, 331 elderly people living alone were registered to receive health services from the district public health center. AD-8 was arranged as one of the health assessment for them. Among them, 62 elderly people who had MCI were eligible participants for this study. The prevalence rate close to that reported in previous studies [26]. We visited them in their homes, one by one, following the order of the roster. Those who were not at home during home visit, did not agree to participate, or could not have time to participate were excluded from this study. 

The inclusion criteria were as follows: (a) 70 years old or older; (b) able to communicate in Chinese and/or Taiwanese; (c) living alone; (d) AD-8 Dementia Screening Interview ≥2; (e) willing to participate in the study and signed the consent form. Finally, 30 older adults were included in the study. Webler, Danielson, and Tuler [27] proposed at least three participants loading on each perspective and, typically, 12 to 36 participants as sufficient for a Q methodology study. 

### 2.3. Q-sorting

We collected information about the participant demographics, including age, gender, marital status, education, and income level, using a questionnaire. The participants gained information on a dementia prevention program by reviewing a brief document that included existing literature suggesting a dementia prevention program [12]. The program included three categories of physical training (e.g., hand-eye coordination training and agility training), brain fitness (e.g., memory training and calculation training), and self-care (e.g., nutritious diet, sleep physiology, and inter-personal skills). The participants were asked to review the program before Q-sorting and were told that this was the dementia prevention program referred to in the Q-statements during the subsequent Q-sorting. Next, participants conducted Q-sorting, subjectively ranking the 32 Q-statements using a 32-cell grid. Each statement was printed on a small card for the participant to read and rank. We asked the participants to rank 32 Q-statements to be placed in a large poster, shown in Figure 1, prepared in advance according to the levels of agreement. We illustrated the operating procedure to participants, provided 32 cards with statements and requested them to review all of the statements carefully. The participants were asked to place the two statements that they agreed with the most (+4, positively labelled) in the far right column and the other two statements they disagreed with the most (−4, negatively labelled) in the far left column. The participants repeated the procedure in the next column (+3 and −3) by selecting three agreed and three disagreed statements from the remaining statements. The procedure was repeated until the Q-sort grids were all completely filled. We allowed the participants to adjust their sorting decisions at any time until the participants did not wish to make any further changes. 

The participants were invited to provide comments while selecting the statements that reflected their own perspectives on participating in a dementia prevention program. Their comments were recorded and extracted to support statements in the results section. The narrative descriptions were extracted from those comments to reveal examples of patterns of perspective.

### 2.4. Ethical Considerations

The study protocol was approved by the institutional review board of the National Taiwan Normal University (IRB approval no. 201803HS018) and all the participants provided written informed consent. 

### 2.5. Data Analysis 

The PQ Method software (Version 2.35) was used to analyze the collected data. Q-sort factor analysis rather than traditional item factor analysis was used to form participant groups (factors), based on similarities in their Q sorts. The major difference between the two types of factor analyses was that the Q-sort factor analysis clusters people instead of clustering question items. Factors were extracted using principal component analysis with Varimax rotation applied to maximize the similarities within factors and the differences between them. Rotation ensured that each participant was loaded on only one factor. The participants who loaded significantly on a factor had similar perspectives about participation in a dementia prevention program. The relative contribution of each factor in explaining the total variance in the data set was determined using eigenvalues. A combination of eigenvalues reflects the amount of variance accounted for by a corresponding factor. Four factors with eigenvalues greater than one were derived. The statistical analysis revealed that the four-factor solution was the best fit for the data. 

## 3. Results

### 3.1. Characteristics of Participants

As shown in Table 2, 50% of the participants were over 76 years old. The participants comprised 22 females and 8 males; 53.3% of the participants had an elementary school education; 43.3% of the participants were from low-income households. 

### 3.2. Factor Analysis

Through the statistical analyses, four factors were extracted that accounted for 54.65% of the total variance. The description of each factor is presented based on the ranked statements at both ends during the Q-sorting process (i.e., +4 and −4). In addition, if the Q-statements ranked as +3 or −3 were distinguishing statements (*p* < 0.05), they were also included to highlight the differences among the extracted factors. Table 3 presents the Q-statements and four factors to reveal the four patterns of perspectives. The four patterns of perspectives include (a) awareness of health hazards and readiness to take preventive actions; (b) emphasis on cost considerations and not being ready to participate; (c) concern about family attitudes and need for family support; (d) emphasis on medical care and needing providers’ recommendation.

### 3.3. Group 1: Awareness of Health Benefits and Readiness to Take Preventive Action (Health Promotion)

Elders who were classified into this perspective pattern recognized the health benefits and were ready to take preventive action. However, the participants might be concerned about physical load when participation is too challenging for them as it may be physically strenuous. According to the characterizing statements, they agreed that participating in a dementia prevention program can improve physical and mental health (Q9, +4), increase knowledge (Q11, +3), and increase physical activity (Q10, +3), but they were concerned that their level of physical load would influence their participation in a dementia prevention program (Q22, +3).

Those who shared this perspective pattern revealed a similar perspective—for example, participants 08, 09, 10, and 27 provided comments such as “participating in multiple cognitive activities will promote health”, “be happy to recommend that neighbors participate in a dementia prevention program to improve health”, “would like to participate in cognitive activities hosted by the community”, and “I will do my best to participate in such a program to improve my health”, respectively. Participant 08 also claimed that he was undergoing rehabilitation treatment after a stroke attack, worrying that “if some of the program activities are strenuous, they are too challenging for me”, so physical load needs to be considered in relation to participation. 

### 3.4. Group 2: Emphasis on Cost Considerations and Not Ready to Participate (Cost Considerations)

Elders who shared this perspective pattern agreed that costs would influence their likelihood of participation in a dementia prevention program. According to the characterizing statements, cost considerations referred to the travel distance (Q14, +4), the program fees (Q15, +3), the total number of sessions (Q16, +3), and the duration of each session (Q17, +3). 

Those who shared this perspective pattern provided similar narratives. For example, participants 05 and 18 both stated that the location of a dementia prevention program should be “close to my residence or living community”. Participant 05 insisted that “the program should be free of charge, otherwise I would not participate in it”, while Participant 19 claimed that “I need to take care of my sick daughter, and cannot leave home for too long to participate in a dementia program”. Participant 25 suggested that “the total number of sections and the duration of each session should be short”. 

### 3.5. Group 3: Concern about Family Attitude and Needing Family Support (Family Ties)

Elders who shared this perspective pattern emphasized family response as a key to participating in the program. They still keep in touch and interact with their families and would not like to be a burden on them. According to the characterizing statements, they strongly agreed they would participate in a dementia prevention program because their children expect them to do so (Q28, +4). They also agreed that their participation depended on the fear of suffering dementia (Q5, +3), the travel distance (Q14, +3), and maintaining independence. 

Those who shared this perspective reveal similar narratives, for example, participants 03, 07, and 30 would participate in a dementia prevention program “at the request of their children”. Participants 03 and 30 expressed that “We still have certain forms of family activities, such as holiday reunions or occasional visits, so we were deeply concerned about problems of poor memory and health that may prevent us from participating in family activities”. Participants 03, 07, and 30 would like to “maintain independence to avoid becoming a burden on their family”. 

### 3.6. Group 4: Emphasis on Medical Care and Needing Providers’ Recommendation (Medical Care)

Participants with this perspective agreed with the importance of medical care and needed their providers’ recommendation. According to the characteristic statements, they agreed that “the medical expenses of suffering dementia would influence my participation in a dementia prevention program” (Q18, +4). They agreed that “participating in a dementia prevention program can maintain my independence” (Q13, +3) and “would like to participate in a dementia prevention program because of the advice of providers” (Q27, +3).

Those who shared this perspective pattern revealed similar narratives. For example, participant 20 emphasized that “I cared about my health and would participate in a dementia prevention program because of my physician’s advice”. 

## 4. Discussion

Based on the 6C framework, this study revealed four perspectives of the elderly on a dementia prevention program. The four patterns of shared perspectives, which accounted for 54.65% of the total variance, included (a) the awareness of health benefits and readiness to take preventive actions; (b) emphasis on cost consideration and not ready to participate; (c) concern about family attitude and needing family support; (d) emphasis on medical care and needing their providers’ advice. Meeting their needs based on older adults’ various perspectives will enable more effective marketing and planning of dementia prevention programs for the elderly in need, increase their willingness to participate, and reduce obstacles to participation, so as to successfully promote the program.

The elderly living alone have multiple risks [28] and a higher relative risk for developing dementia [9]. Moreover, elderly people with MCI are more likely to develop dementia [29]. Our participants are living alone with identified MCI; thus, they are one of the top priority at-risk subpopulations for dementia prevention. Especially in an aged society, whether they are willing to participate in a dementia prevention program is critical for dementia prevention. We can more confidently infer that the views represent the participants’ experiences because of the forced distribution of Q-sorts and discussions in subsequent interviews. The findings contribute to the literature by revealing four prominent perspectives and marketing strategies for a dementia prevention program. 

The first perspective group of elderly agreed most that a dementia prevention program can improve physical and mental health, improve physical activity, and increase their own knowledge, and that their own physical load would also influence their participation in the program. They give priority to attending to their own health and self-needs. Corresponding to the 6C marketing mix, they are classified as the pattern of **consumers** who are concerned about health benefits and barriers (ability to afford program requirements). A previous study [30] indicated that the most critical consideration of the elderly is their health, and the most worrying problem is their health deterioration. A significant obstacle to participating in preventive services is the elder’s inability to deal with their physical load [31]. 

The elderly people who fit into perspective pattern two agreed that it is critical to spend time and pay the cost of participating in a dementia prevention program, even if they already suffer MCI. Corresponding to the 6C framework, this pattern of elderly pay attention to the **cost and convenience** involved in participating in the program. A study which investigated the needs of attending a fall prevention program among older people, also revealed that time and cost associated with program participation should be taken into account in addition to relevant content (such as knowledge/strategies of fall prevention) [32]. Program designers should be aware of the need to provide a low-cost and convenient dementia prevention program for elderly people to engage in preventive practices. 

The elders who fit into perspective pattern three emphasized the burden on their children if they developed dementia and that their children’s encouragement would shape their willingness to participate. The desire to be a considerate parent was revealed as a motivation to protect children from becoming caregivers of a dementia patient. A previous finding indicated that older adults believed that health problems caused by falling were serious, and a fall prevention program could decrease the burden they place on their family [17]. Our findings were consistent with that study in that the elderly in perspective pattern three believed that participating in a dementia prevention program would help them to avoid suffering dementia. This corresponds to the 6C marketing mix that identifies the **community** (interpersonal impact) and **concern** (aging impact) elements. Compared to the second perspective group, this group of elderly did not care about the cost of participating in the dementia prevention program. 

Elders who fit in with perspective pattern four shared the view of adhering to their providers’ advice. They have high trust and compliance with their providers, and are willing to take advice from them regarding dementia prevention. In addition, they agree that if they suffer dementia, it will incur subsequent medical costs, so they are willing to participate in a dementia prevention program. Corresponding to the 6C marketing mix, these elderly are associated with the elements of **communication and cost**. Providers’ advice plays a critical role in initial receipt of a preventive health service, which is supported by the findings of a prior study [33]. The implication for practitioners is that marketing of the dementia prevention program should include the provider as an important partner to promote the participation of the elderly.

## 5. Study Implementations and Limitations

Our findings increase the current understanding of the distinct perspectives of elderly people with MCI living alone regarding participation in a dementia prevention program. The participants also expressed comments associated with their Q sorts that enhance the appropriate interpretation of the study findings. Our findings could be beneficial in improving the effectiveness of interventions targeting older adults to reduce their risk of suffering dementia. Health professionals could apply appropriate marketing strategies to encourage older adults to engage in dementia prevention. Responsive approaches based on older adults’ perspective patterns would be beneficial for attaining more satisfactory dementia prevention. To our best knowledge, this is the first study to identify the perspective patterns associated with older adults with mild MCI, which could further inform suitable strategies and sustainable applications.

The current study could not avoid some limitations. Including a comparison of gender differences would have advanced the understanding of similarities and differences between females and males in participating a dementia program. However, there were only eight male participants in the current study, and such a comparison could not be conducted. Gender comparison is encouraged in studies with sufficient sample size. Further, our study considered only community-dwelling, rather in-facility, participants. Older adults living in long-term care facilities may have different perspectives regarding their participation in a dementia prevention program. Hence, we suggest that future studies conduct a Q methodology study with a sample of older adults that includes those living in long-term care facilities.

## 6. Conclusions

Our findings revealed the shared perspectives of four groups of community-dwelling elderly with MCI living alone. The exploration of clusters of the elderly with MCI could assist health professionals in acknowledging older people’s characteristic attitudes and responses towards participation in a dementia prevention program. These findings could serve as a useful guide for researchers and practitioners to develop tailored dementia-prevention programs targeting elderly people with MCI living alone.

## Figures and Tables

**Figure 1 ijerph-17-07712-f001:**
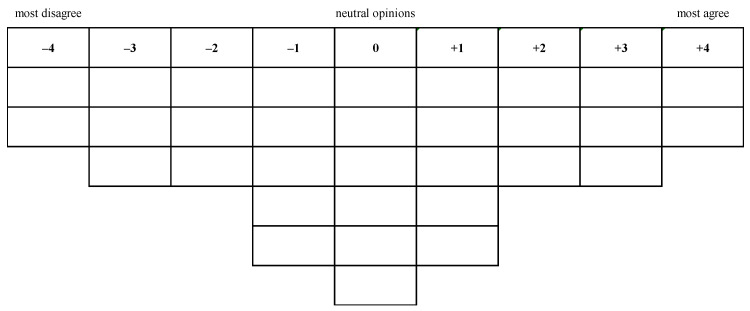
Forced-choice frequency distribution in the Q-sorting process.

**Table 1 ijerph-17-07712-t001:** Interview questions.

Interview Questions
Are you concerned about dementia? Why? What are the reasons causing your concern about dementia? (concern) Have you ever learned about dementia prevention? If you attend a dementia prevention program, what are the possible advantages? (consumer) If you have an opportunity to attend a dementia prevention program, what factors may decrease your interest in attending the program or prevent you from participating? (cost) If you have an opportunity to attend a dementia prevention program, what strategies or methods may make it more convenient for you to attend the program? (convenience) Where would you find information on attending a dementia program? Will the information affect your intention to attend? (communication) Who may influence your interest in attending a dementia prevention program? (community)

**Table 2 ijerph-17-07712-t002:** Characteristics of the participants.

Variable	All (N = 30)
Age, *n* (%)	
under 75 years old	15 (50.0%)
76–80 years old	11 (36.7%)
over 81 years old	4 (13.3%)
Gender, *n* (%)	
women	22 (73.3%)
men	8 (26.7%)
Marital status, *n* (%)	
ever married	22 (73.3%)
never married	8 (26.7%)
Education level, *n* (%)	
below elementary school	16 (53.3%)
junior or high school	9 (30.0%)
university (college) or above	5 (16.7%)
Economic level, *n* (%)	
not receiving financial assistance	17 (56.7%)
received subsidies from low-income or middle-income	13 (43.3%)

**Table 3 ijerph-17-07712-t003:** Q-statements and factor arrays across the four factors.

Q-Statement	F1	F2	F3	F4
	N = 11	N = 6	N = 5	N = 2
Attitudes toward participating in a dementia prevention program				
Concern				
1. I worry about memory decline.	0	−2	2	−2
2. I had the experience of a blank brain.	−1	0	−1	1
3. I had experiences of failing to convey meaning and difficulty in expressing myself.	−2	−1	0	−1
4. I worry that if I suffer from dementia, my family will need to take care of me.	1	−1	2	0
5. I am afraid of suffering dementia because I am getting older.	1	−2	**3**	2
6. I would like to participate in a dementia prevention program because I have risk factors for dementia (e.g., fall, high blood pressure, high cholesterol, high blood sugar, depression, and inadequate social interaction).	−2	0	0	−4
Consumer				
7. I think participating in a dementia prevention program can help me find new friends.	1	1	−1	0
8. I think by participating in a dementia prevention program, I can adequately interact with friends.	2	−1	−2	2
9. I think participating in a dementia prevention program can improve physical and mental health.	**4**	0	1	−2
10. I think participating in a dementia prevention program can increase physical activity.	**3**	1	1	1
11. I think participating in a dementia prevention program can increase knowledge.	**3**	−3	**3**	**3**
12. I think participating in a dementia prevention program can be a good use of my time.	1	0	0	2
13. I think participating in a dementia prevention program can maintain my independence.	2	1	−1	**3**
Cost				
14. The distance to travel will influence my participation in a dementia prevention program.	2	**4**	**3**	1
15. The tuition fees will influence my participation in a dementia prevention program.	2	3	−2	−4
16. The total number of sessions will influence my participation in a dementia prevention program.	−1	**3**	0	0
17. The duration of each session will influence my participation in a dementia prevention program.	0	**3**	−2	−1
18. The medical expenses of suffering dementia will influence my participation in a dementia prevention program.	−1	1	1	**4**
Convenience				
19. The face-to-face teaching methods will influence my participation in a dementia prevention program.	−3	2	−1	−3
20. The online teaching methods will influence my participation in a dementia prevention program.	−3	2	0	2
21. The learning difficulty of the program will influence my participation in a dementia prevention program.	0	2	−2	−3
22. The level of physical load will influence my participation in a dementia prevention program.	**3**	2	2	0
Communication				
23. I would like to participate in a dementia prevention program because I have learned information about dementia.	−2	−1	−1	1
24. I would like to participate in a dementia prevention program because of a friend’s suggestion.	0	−3	−3	0
25. I would like to participate in a dementia prevention program because of the influence of social media.	−4	−4	−4	−1
26. I would like to participate in a dementia prevention program because of an introduction by the staff of the elderly service center.	−2	−3	−3	1
27. I would like to participate in a dementia prevention program because of the advice of providers.	−1	−1	0	**3**
Community				
28. I would like to participate in a dementia prevention program because my children expect me to come to class.	−3	−2	**4**	−2
29. I would like to participate in a dementia prevention program because of a friend’s invitation.	0	−2	−3	0
30. I would like to participate in a dementia prevention program because I have actually contacted a dementia patient.	0	0	2	−1
31. I would like to participate in a dementia prevention program because some family or friend suffers from dementia and I would not like to have dementia like them.	−1	1	1	−1
32. I would like to participate in a dementia prevention program because I would not like to become a burden for significant others.	1	0	1	−2

Note: The numbers **4** and **3** represent that the statements most accurately reflected the experience of participants who loaded significantly onto the given group.

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
