# Peer review of "Perspectives of the Elderly with Mild Cognitive Impairment Living Alone on Participating in a Dementia Prevention Program: A Q Methodology Study"

_ijerph, 2020, doi:10.3390/ijerph17217712_

Round 1
Reviewer 1 Report
Page 4, lines 139-149: You state the exclusion and inclusion criteria. Could you give information on who were excluded? I mean how many were excluded because they were not willing and how many because they did not meet the inclusion criteria? Also, how did you know in advance that 62 persons of 331 had MCI? Did you perform the AD-8 Dementia Screening Interview as part of the home visit or how did you obtain the score?
Page 5, lines 181-182: Is there something missing here? Or what do you mean by “Factor analysis rather than traditional item factor analysis was used to…” . Could you just say that you used factor analysis?
Page 10, lines 319-328: This text is more about study implications than strengths.
Author Response
Authors’ response
Review (1)
Page 4, lines 139-149: You state the exclusion and inclusion criteria. Could you give information on who were excluded? I mean how many were excluded because they were not willing and how many because they did not meet the inclusion criteria? Also, how did you know in advance that 62 persons of 331 had MCI? Did you perform the AD-8 Dementia Screening Interview as part of the home visit or how did you obtain the score?
Author response
The 331 elderly were registered to receive health services from a district public health center. AD-8 was arranged as one of the health assessment for them. Among them, 62 elderly who had MCI were eligible participants of this study. However, some of the elderly with MCI were not at home during home visit, did not agree to participate, or could not have time to participate. Finally, 30 older adults were included in the study. Please see page 4.
Page 5, lines 181-182: Is there something missing here? Or what do you mean by “Factor analysis rather than traditional item factor analysis was used to…” . Could you just say that you used factor analysis?
Author response
The difference between Q-sort and traditional item factor analyses were described as followed: Q-sort factor analysis rather than traditional item factor analysis was used to form participant groups (factors), based on similarities in their Q sorts. The major difference between the two types of factor analyses was that Q-sort factor analysis cluster person instead of cluster question items. Please see page 5.
Page 10, lines 319-328: This text is more about study implications than strengths.
Author response
The subtitle was revised as suggested. Please see page 10.

Reviewer 2 Report
The manuscript has improved and I believe now it may be ready for publication. My only suggestion (and the authors do not need to necessarily address) remains in the discussion by explaining in which ways individuals living in log term care residences would report differing views on prevention programs compared to community residents (Lines 334-337).
No need for me to look at next draft.
Author Response
Authors’ response
Review (2)
The manuscript has improved and I believe now it may be ready for publication. My only suggestion (and the authors do not need to necessarily address) remains in the discussion by explaining in which ways individuals living in long term care residences would report differing views on prevention programs compared to community residents (Lines 334-337).
No need for me to look at next draft.
Author response
Thank you for the common. Indeed, long term care residences may have different view on prevention programs. We would like to explore their viewpoints in depth, and it is also a direction of study in the future.

This manuscript is a resubmission of an earlier submission. The following is a list of the peer review reports and author responses from that submission.
Round 1
Reviewer 1 Report
Thank you for this good reading. Please see below for my comments/suggestions:
Thank you for this manuscript, I appreciate the novelty in the use of 4C/6C but I am afraid I am entirely clear how the methodology has been applied for your purposes. I suggest addressing my comments, hoping that you will find them useful.
Abstract:
- Lines 19-21 were these Q-statements defined a priori or ensuing from the analysis? I would think they would ensue from the analysis but please clarify.
- Line 20-21 'Participants performed Q22 sorting to rank the statements into a Q-sort grid'. Can you please rephrase this more clearly?
- Lines 27-28 this is not what your findings reveal, I would suggest to draw directly form your findings as the sentence currently is a bit vague.
Introduction:
- Line 35 I would insert the phrasing '...public health initiatives' in the sentence: 'With the development of medical technology and public health,
- Line 43 I suggest the use of 'may' in the sentence: '...a health crisis will result'.
- Line 46 should be 'statistically significantly associated...'
- Lines 71-74 can be deleted.
- Line 84 is this (i.e. the 4C) your framework? You need to explain better what it is and why it is important in the context of your investigation.
- Lines 94-96 Not sure how 4C became 6C and not sure its relevance in the study.
- Lines 97-114 sounds a bit like a long list which is not tailored to the study. I think that if 6C is your framework informing the analysis for your findings you need to tailor it according to the context and objectives of your study.
- Lines 15-16: 'The Q methodology can identify the participants’ heterogeneity of subjectivity through the process of The Q methodology can identify the participants’ heterogeneity of subjectivity through the' What you mean in this sentence? It is not clear.
Methods:
- Lines 139: Not sure you need to include these references about sample extimation if then you have not followed their examples.
- Lines 145-155 need rephrasing as read proof.
- Ethics: do you need to include a code/number of ethical approval?
- If you use stats you may need to mention this in the methods.
Results:
- This is much clearer but difficult to follow.
- most comments lack detailed information on participant number (e.g. participant 05 etc...)please check all results. Also in some themes (e.g. 3.3 Group 2: Emphasis on Cost Considerations and not Being Ready to Participate, there are no participants' quotations to back up your assertion.
Discussion:
I am not sure I have a clear picture of the study, I see what you are trying to convey but it is difficult to follow. I think a new version need to pay attention to the way methods-results-discussion are linked to one another.
Author Response
Abstract:
Lines 19-21 were these Q-statements defined a priori or ensuing from the analysis? I would think they would ensue from the analysis but please clarify.
Response: The process through which the 32 statements were derived is fully explained on page 3 of the revised manuscript.
Line 20-21 'Participants performed Q-sorting to rank the statements into a Q-sort grid'. Can you please rephrase this more clearly?
Response: We revised the sentence so that the Q-sort procedure is now fully explained on page 4.
Lines 27-28 this is not what your findings reveal, I would suggest to draw directly from your findings as the sentence currently is a bit vague.
Response: We revised the sentence as follows: As Q methodology applies a forced distribution through the Q‐sorting technique, it could capture participants’ perspective patterns. (p. 1)
Introduction:
Line 35 I would insert the phrasing '...public health initiatives' in the sentence: 'With the development of medical technology and public health,
Response: We revised the text as suggested.
Line 43 I suggest the use of 'may' in the sentence: '...a health crisis will result'.
Response: We revised the text as suggested.
Line 46 should be 'statistically significantly associated...'
Response: We revised the text as suggested.
Lines 71-74 can be deleted.
Response: We deleted the text as suggested.
Line 84 is this (i.e. the 4Cs) your framework? You need to explain better what it is and why it is important in the context of your investigation.
Response: We expanded the explanation to clarify the importance of the 4Cs to our framework (pp. 2-3).
Lines 94-96 Not sure how 4Cs became 6Cs and not sure its relevance in the study.
Response: We clarified the modification of the 4Cs to become the 6Cs, and stated the relevance of 6Cs for our study (pp. 2-3).
Lines 97-114 sounds a bit like a long list which is not tailored to the study. I think that if 6Cs is your framework informing the analysis for your findings you need to tailor it according to the context and objectives of your study.
Response: We clarified our use of the 6Cs as the study framework. In terms of the research process (p. 3), we first used 6Cs to formulate interview questions (Table 1). After the interview, 162 statements were extracted from the verbatim interview transcripts. Then, the research team conducted a thorough review and condensed them into 32 statements. Three health professionals were invited to examine content validity.
Lines 115-16: 'The Q methodology can identify the participants’ heterogeneity of subjectivity through the process of “Q sorting”. What you mean in this sentence? It is not clear.
Response: We elaborated on the Q-sorting process on page 4.
Methods:
Lines 139: Not sure you need to include these references about sample estimation if then you have not followed their examples.
Response: We clarified the sampling process in the revised manuscript (p. 4).
Lines 145-155 need rephrasing as read proof.
Response: We rephrased the text as suggested. We also attached an editing certificate from a professional English editing service.
Ethics: do you need to include a code/number of ethical approval?
Response: We added the IRB approval no. 201803HS018 on page 5 of the revised manuscript.
If you use stats you may need to mention this in the methods.
Response: We used factor analysis, as discussed on page 5 of the revised manuscript.
Results:
This is much clearer but difficult to follow.
Response: We added the subtitle “Factor analysis” for improved clarity, and the manuscript was revised for improved clarity, readability, and logical flow..
most comments lack detailed information on participant number (e.g. participant 05 etc...) please check all results. Also in some themes (e.g. 3.3 Group 2: Emphasis on Cost Considerations and not Being Ready to Participate, there are no participants' quotations to back up your assertion.
Response: We added participant numbers and quotations to back up our assertion in the results section, as suggested.
Discussion:
I am not sure I have a clear picture of the study, I see what you are trying to convey but it is difficult to follow. I think a new version need to pay attention to the way methods-results-discussion are linked to one another.
Response: We revised the methods, results, and discussion sections to clarify the linkages between them.
Reviewer 2 Report
Review report IJERPH 898994
This study examined older persons’ attitudes about a dementia prevention program using Q methodology. This is an interesting and promising approach and it has the potential to provide novel information. However, the study concept and aim is not clearly presented. I am not sure whether this study aims to find the best way to develop and market a dementia prevention program, or if the aim is to present older persons’ perspectives about an ongoing program that they are participating or are about to participate. The introduction includes information on previously implemented programs and marketing strategies, however a clarification of the study design is lacking. I present my concerns in the comments below.
Introduction:
Lines 52-53: Why do Asian people need to pay more attention to dementia than those living in Europe and America? I am not convinced about this conclusion. Should they pay attention to the potential symptoms earlier?
Line 58: What does life function mean?
Line 60: Computing power, do you mean numeracy?
Method:
Lines 130-143: Were the participants already enrolled in and participating in a dementia prevention program, were they about to start one, or was the idea to ask participants perspectives about a potential prevention program that might take place in the future? This is not clearly stated, but it would be very important information for the reader.
Line 135-135: If you selected participants by visiting the potential participants in the order of the roster, does this potentially cause selection bias? This should be discussed in the study limitations.
Lines 151-152, 145-146: The Q statements include the term “dementia prevention program”. What were the participants told about this program? Was it explained in the interview questions or how did the participants know what they were asked about? Was there a model program? Please clarify this.
Results
Lines 208-209: What do you mean by ”physical load”? Do you mean that they have physical limitations in functioning, which can compromise their participation? Or was participation too challenging for them, because it was physically too strenuous? Please clarify this.
Line 244: maintain independence?
Discussion
In my opinion, it would be better to start the discussion with your results , rather than summarizing findings form other studies.
Line 259: Please provide descriptive words for social relations and economic status (such as adequate number of social relations and good economic status), to have it in line with other statements.
How does the chosen approach relate to other strategies to evaluate attitudes about participation among older persons with MCI? The references that you give are not from dementia prevention programs.
Strengths and limitations are missing.
There are some typos and minor grammatical errors, please proof-read the text.
Author Response
This study examined older persons’ attitudes about a dementia prevention program using Q methodology. This is an interesting and promising approach and it has the potential to provide novel information. However, the study concept and aim is not clearly presented. I am not sure whether this study aims to find the best way to develop and market a dementia prevention program, or if the aim is to present older persons’ perspectives about an ongoing program that they are participating or are about to participate. The introduction includes information on previously implemented programs and marketing strategies, however a clarification of the study design is lacking. I present my concerns in the comments below.
Response: We appreciate the comment. The study aimed to clarify older adults’ perspectives on a dementia prevention program in which they were about to participate. Referring to the Japanese dementia prevention program mentioned in manuscript [12], the research team adjusted the program to meet the needs of Taiwanese older adults at risk of dementia. In order to elaborate on the attitudes of elderly with MCI regarding their participation in this dementia prevention program, and to effectively market the program to them, we conducted this study using in-depth interviews to develop Q-statements, and Q methodology to explore older adults’ perspectives of participation in a dementia program.
Introduction:
Lines 52-53: Why do Asian people need to pay more attention to dementia than those living in Europe and America? I am not convinced about this conclusion. Should they pay attention to the potential symptoms earlier?
Response: We revised the sentence as follows, for improved clarity:
Thus, older adults in Asia need to pay attention to dementia earlier in their lifespan, compared to their counterparts in Europe and the US. (p. 2)
Line 58: What does life function mean?
Response: Reduced life function refers to having an increasing need for assisted self-care. We revised the term on page 2 to improve clarity.
Line 60: Computing power, do you mean numeracy?
Response: We revised the text as suggested.
Method:
Lines 130-143: Were the participants already enrolled in and participating in a dementia prevention program, were they about to start one, or was the idea to ask participants perspectives about a potential prevention program that might take place in the future? This is not clearly stated, but it would be very important information for the reader.
Response: Participants were informed of a dementia prevention program by reviewing a brief document that included existing literature [12] related to a dementia prevention program. In order to elaborate on the attitudes of the elderly with MCI regarding participation in the program, and to effectively market the program to them, we conducted this study using in-depth interviews and Q methodology.
Line 135-135: If you selected participants by visiting the potential participants in the order of the roster, does this potentially cause selection bias? This should be discussed in the study limitations.
Response: We appreciate the comment. We revised our description of the sampling procedure to clarify the recruitment process (p. 4).
Lines 151-152, 145-146: The Q statements include the term “dementia prevention program”. What were the participants told about this program? Was it explained in the interview questions or how did the participants know what they were asked about? Was there a model program? Please clarify this.
Response: Participants were informed of a dementia prevention program by reviewing a brief document that included existing literature [12] related to a dementia prevention program [12]. The program included three categories: physical training (i.e. hand eye coordination training, agility training, etc.), brain fitness (i.e. memory training, calculation training, etc.), and self-care (i.e. nutritive diet, sleep physiology, inter personal skills, etc.). Participants were asked to review the dementia prevention program before Q-sorting, and told that this was the program referred to in the Q statements during the subsequent Q-sorting phase (p. 4).
Results
Lines 208-209: What do you mean by ”physical load”? Do you mean that they have physical limitations in functioning, which can compromise their participation? Or was participation too challenging for them, because it was physically too strenuous? Please clarify this.
Response: Physical load refers to participation becoming too challenging for participants, e.g. being physically too strenuous for vulnerable participants. We revised the text to clarify the meaning (p. 6).
Line 244: maintain independence?
Response: We revised the text as suggested.
Discussion
In my opinion, it would be better to start the discussion with your results, rather than summarizing findings form other studies.
Line 259: Please provide descriptive words for social relations and economic status (such as adequate number of social relations and good economic status), to have it in line with other statements.
Response: We revised the paragraph and deleted the sentences from line 257 to line 261, as social relations and economic status are not related to the study findings.
How does the chosen approach relate to other strategies to evaluate attitudes about participation among older persons with MCI? The references that you give are not from dementia prevention programs.
Response: We revised the text as follows (p. 9):
A study which investigated the needs of attending a fall prevention program among older people, also revealed that time and cost associated with program participation should be taken into account in addition to relevant content (such as knowledge/strategies of fall prevention) [33].
Strengths and limitations are missing.
Response: We included strengths and limitations in the revised manuscript (p. 10).
There are some typos and minor grammatical errors, please proof-read the text.
Response: We attached an editing certificate from a professional English editing service.